# NEURAL MODULE NETWORKS FOR REASONING OVER TEXT

**Nitish Gupta[1], Kevin Lin[2]\*, Dan Roth[1], Sameer Singh[3] & Matt Gardner[4]**
{nitishg,danroth}@seas.upenn.edu, kevinlin@eecs.berkeley.edu,
sameer@uci.edu, mattg@allenai.org
[1]University of Pennsylvania, Philadelphia, [2]University of California, Berkeley,
[3]University of California, Irvine, [4]Allen Institute for AI

## ABSTRACT

Answering compositional questions that require multiple steps of reasoning against text is challenging, especially when they involve discrete, symbolic operations. Neural module networks (NMNs) learn to parse such questions as executable programs composed of learnable modules, performing well on synthetic visual QA domains. However, we find that it is challenging to learn these models for non-synthetic questions on open-domain text, where a model needs to deal with the diversity of natural language and perform a broader range of reasoning. We extend NMNs by: (a) introducing modules that reason over a paragraph of text, performing symbolic reasoning (such as arithmetic, sorting, counting) over numbers and dates in a probabilistic and differentiable manner; and (b) proposing an unsupervised auxiliary loss to help extract arguments associated with the events in text. Additionally, we show that a limited amount of heuristically-obtained question program and intermediate module output supervision provides sufficient inductive bias for accurate learning. Our proposed model significantly outperforms state-of-the-art models on a subset of the DROP dataset that poses a variety of reasoning challenges that are covered by our modules.

## 1 INTRODUCTION

Being formalism-free and close to an end-user task, QA is increasingly becoming a proxy for gauging a model's natural language understanding capability (He et al., 2015; Talmor et al., 2018). Recent models have performed well on certain QA datasets, sometimes rivaling humans (Zhang et al., 2019), but it has become increasingly clear that they primarily exploit surface level lexical cues (Jia & Liang, 2017; Feng et al., 2018) and compositional QA still remains a challenge. Answering complex compositional questions against text is challenging since it requires a comprehensive understanding of both the question semantics and the text against which the question needs to be answered. Consider the question in Figure 1; a model needs to understand the compositional reasoning structure of the questions, perform accurate information extraction from the passage (eg. extract *lengths*, *kickers*, etc. for the *field goals* and *touchdowns*), and perform symbolic reasoning (eg. counting, sorting, etc.).

Semantic parsing techniques, which map natural language utterances to executable programs, have been used for compositional question understanding for a long time (Zelle & Mooney, 1996; Zettlemoyer & Collins, 2005; Liang et al., 2011), but have been limited to answering questions against structured and semi-structured knowledge sources. Neural module networks (NMNs; Andreas et al., 2016) extend semantic parsers by making the program executor a *learned function* composed of neural network modules. These modules are designed to perform basic reasoning tasks and can be composed to perform complex reasoning over unstructured knowledge.

NMNs perform well on synthetic visual question answering (VQA) domains such as CLEVR (Johnson et al., 2017) and it is appealing to apply them to answer questions over text due to their interpretable, modular, and inherently compositional nature. We find, however, that it is non-trivial to extend NMNs for answering non-synthetic questions against open-domain text, where a model needs to deal with

---

\*Work done while at Allen Institute for AI.

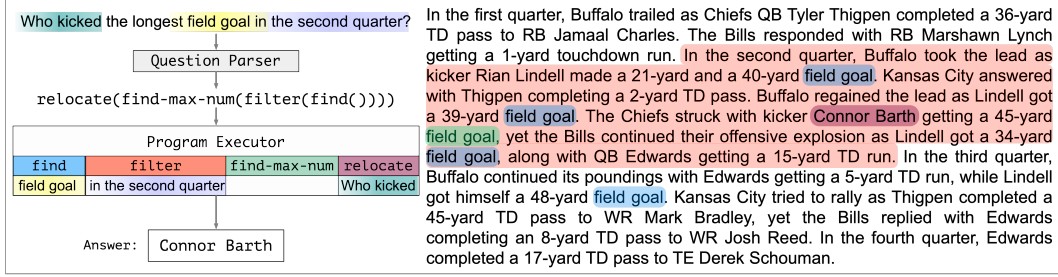

Figure 1: **Model Overview**: Given a question, our model parses it into a program composed of neural modules. This program is executed against the context to compute the final answer. The modules operate over soft attention values (on the question, passage, numbers, and dates). For example, `filter` takes as input attention over the question (*in the second quarter*) and filters the output of the `find` module by producing an attention mask over tokens that belong to the *second quarter*.

the ambiguity and variability of real-world text while performing a diverse range of reasoning. Jointly learning the parser and executor using only QA supervision is also extremely challenging (§2.2).

Our contributions are two-fold: Firstly, we extend NMNs to answer compositional questions against a paragraph of text as context. We introduce neural modules to perform reasoning over text using distributed representations, and perform symbolic reasoning, such as arithmetic, sorting, comparisons, and counting (§3). The modules we define are probabilistic and differentiable, which lets us maintain uncertainty about intermediate decisions and train the entire model via end-to-end differentiability.

Secondly, we show that the challenges arising in learning from end-task QA supervision can be alleviated with an auxiliary loss over the intermediate latent decisions of the model. Specifically, we introduce an unsupervised objective that provides an inductive bias to perform accurate information extraction from the context (§4.1). Additionally, we show that providing heuristically-obtained supervision for question programs and outputs for intermediate modules in a program (§4.2) for a small subset of the training data (5–10%) is sufficient for accurate learning.

We experiment on 21,800 questions from the recently proposed DROP dataset (Dua et al., 2019) that are heuristically chosen based on their first n-gram such that they are covered by our designed modules. This is a significantly-sized subset that poses a wide variety of reasoning challenges and allows for controlled development and testing of models. We show that our model, which has interpretable intermediate outputs by design, significantly outperforms state-of-the-art black box models on this dataset. We conclude with a discussion of the challenges of pushing NMNs to the entire DROP dataset, where some questions require reasoning that is hard to design modules for.

## 2 Neural Module Networks

Consider the question *"Who kicked the longest field goal in the second quarter?"* in Figure 1. Multiple reasoning steps are needed to answer such a question: find all instances of "field goal" in the paragraph, select the ones "in the second quarter", find their lengths, compute the "longest" of them, and then find "who kicked" it. We would like to develop machine reading models that are capable of understanding the context and the compositional semantics of such complex questions in order to provide the correct answer, ideally while also explaining the reasoning that led to that answer.

Neural module networks (NMN) capture this intuition naturally, which makes them a good fit to solve reasoning problems like these. A NMN would parse such a question into an executable program, such as `relocate(find-max-num(filter(find())))`, whose execution against the given paragraph yields the correct answer. These programs capture the abstract compositional reasoning structure required to answer the question correctly and are composed of learnable modules designed to solve sufficiently independent reasoning tasks. For example, the `find` module should ground the question span "field goal" to its various occurrences in the paragraph; the module `find-max-num` should output the span amongst its input that is associated with the largest length; and finally, the `relocate` module should find "who kicked" the *field goal* corresponding to its input span.

## 2.1 Components of a NMN for Text

**Modules.** To perform natural language and symbolic reasoning over different types of information, such as text, numbers, and dates, we define a diverse set of differentiable modules to operate over these different data types. We describe these modules and the data types in §3.

**Contextual Token Representations.** Our model represents the question $q$ as $\mathbf{Q} \in \mathbb{R}^{n \times d}$ and the context paragraph $p$ as $\mathbf{P} \in \mathbb{R}^{m \times d}$ using contextualized token embeddings. These are outputs of either the same bidirectional-GRU or a pre-trained BERT (Devlin et al., 2019) model. Here $n$ and $m$ are the number of tokens in the question and the paragraph, respectively. Appendix §A.1 contains details about how these contextual embeddings are produced.

**Question Parser.** We use an encoder-decoder model with attention to map the question into an executable program. Similar to N2NMN (Hu et al., 2017), at each timestep of decoding, the attention that the parser puts on the question is available as a side argument to the module produced at that timestep during execution. This lets the modules have access to question information without making hard decisions about which question words to put into the program.

In our model, the data types of the inputs and output of modules automatically induce a type-constrained grammar which lends itself to top-down grammar-constrained decoding as performed by Krishnamurthy et al. (2017). This ensures that the decoder always produces well-typed programs. For example, if a module $f_1$ inputs a *number*, and $f_2$ outputs a *date*, then $f_1(f_2)$ is invalid and would not be explored while decoding. For example, if a module $f_1$ inputs a *number*, and $f_2$ outputs a *date*, then $f_1(f_2)$ is invalid and would not be explored while decoding. The output of the decoder is a linearized abstract syntax tree (in an in-order traversal). See §A.2 for details.

**Learning.** We define our model probabilistically, i.e., for any given program $\mathbf{z}$, we can compute the likelihood of the gold-answer $p(y^*|\mathbf{z})$. Combined with the likelihood of the program under the question-parser model $p(\mathbf{z}|q)$, we can maximize the marginal likelihood of the answer by enumerating all possible programs; $J = \sum_{\mathbf{z}} p(y^*|\mathbf{z})p(\mathbf{z}|q)$. Since the space of all programs is intractable, we run beam search to enumerate top-K programs and maximize the approximate marginal-likelihood.

## 2.2 Learning Challenges in NMN for Text

As mentioned above, the question parser and the program executor both contain learnable parameters. Each of them is challenging to learn in its own right and joint training further exacerbates the situation.

**Question Parser.** Our model needs to parse free-form real-world questions into the correct program structure and identify its arguments (e.g. "who kicked", "field goal", etc.). This is challenging since the questions are not generated from a small fixed grammar (unlike CLEVR), involve lexical variability, and have no program supervision. Additionally, many incorrect programs can yield the same correct answer thus training the question parser to highly score incorrect interpretations.

**Program Executor.** The output of each intermediate module in the program is a latent decision by the model since the only feedback available is for the final output of the program. The absence of any direct feedback to the intermediate modules complicates learning since the errors of one module would be passed on to the next. Differentiable modules that propagate uncertainties in intermediate decisions help here, such as attention on pixels in CLEVR, but do not fully solve the learning challenges.

**Joint Learning.** Jointly training the parser and executor increases the latent choices available to the model by many folds while the only supervision available is the gold answer. Additionally, joint learning is challenging as prediction errors from one component lead to incorrect training of the other. E.g., if the parser predicts the program `relocate(find())` for the question in Fig. 1, then the associated modules would be incorrectly trained to predict the gold answer. On the next iteration, incorrect program execution would provide the wrong feedback to the question parser and lead to its incorrect training, and learning fails.

# 3 MODULES FOR REASONING OVER TEXT

Modules are designed to perform basic independent reasoning tasks and form the basis of the compositional reasoning that the model is capable of. We identify a set of tasks that need to be performed to support diverse enough reasoning capabilities over text, numbers, and dates, and define modules accordingly. Since the module parameters will be learned jointly with the rest of the model, we would like the modules to maintain uncertainties about their decisions and propagate them through the decision making layers via end-to-end differentiability. One of the main contributions of our work is introducing differentiable modules that perform reasoning over text and symbols in a probabilistic manner. Table 1 gives an overview of representative modules and §3.2 describes them in detail.

| Module | In | Out | Task |
|---|---|---|---|
| `find` | Q | P | For question spans in the input, find similar spans in the passage |
| `filter` | Q, P | P | Based on the question, select a subset of spans from the input |
| `relocate` | Q, P | P | Find the argument asked for in the question for input paragraph spans |
| `find-num` | P | N | } Find the number(s) / date(s) associated to the input paragraph spans |
| `find-date` | P | D | |
| `count` | P | C | Count the number of input passage spans |
| `compare-num-lt` | P, P | P | Output the span associated with the smaller number. |
| `time-diff` | P, P | TD | Difference between the dates associated with the paragraph spans |
| `find-max-num` | P | P | Select the span that is associated with the largest number |
| `span` | P | S | Identify a contiguous span from the attended tokens |

Table 1: Description of the modules we define and their expected behaviour. All inputs and outputs are represented as distributions over tokens, numbers, and dates as described in §3.1.

## 3.1 DATA TYPES

The modules operate over the following data types. Each data type represents its underlying value as a normalized distribution over the relevant support.

- **Question (Q) and Paragraph (P) attentions:** soft subsets of relevant tokens in the text.
- **Number (N) and Date (D):** soft subset of unique numbers and dates from the passage. [1]
- **Count Number (C):** count value as a distribution over the supported count values $(0 - 9)$.
- **Time Delta (TD):** a value amongst all possible unique differences between dates in the paragraph. In this work, we consider differences in terms of years.
- **Span (S):** span-type answers as two probability values (start/end) for each paragraph token.

## 3.2 NEURAL MODULES FOR QUESTION ANSWERING

The question and paragraph contextualized embeddings ($\mathbf{Q}$ and $\mathbf{P}$) are available as global variables to all modules in the program. The question attention computed by the decoder during the timestep the module was produced is also available to the module as a side argument, as described in §2.1.

**find(Q) $\rightarrow$ P**  This module is used to ground attended question tokens to similar tokens in the paragraph (e.g., "field goal" in Figure 1). We use a question-to-paragraph attention matrix $\mathbf{A} \in \mathbb{R}^{n \times m}$ whose $i$-th row is the distribution of similarity over the paragraph tokens for the $i$-th question token. The output is an *expected* paragraph attention; a weighted-sum of the rows of $\mathbf{A}$, weighed by the input question attention, $P = \sum_i Q_i \cdot \mathbf{A}_{i:} \in \mathbb{R}^m$. $\mathbf{A}$ is computed by normalizing (using softmax) the rows of a question-to-paragraph similarity matrix $\mathbf{S} \in \mathbb{R}^{n \times m}$. Here $\mathbf{S}_{ij}$ is the similarity between the contextual embeddings of the $i$-th question token and the $j$-th paragraph token computed as, $\mathbf{S}_{ij} = \mathbf{w_f}^T [\mathbf{Q}_{i:} ; \mathbf{P}_{j:} ; \mathbf{Q}_{i:} \circ \mathbf{P}_{j:}]$, where $\mathbf{w_f} \in \mathbb{R}^{3d}$ is a learnable parameter vector of this module, $[;]$ denotes the concatenation operation, and $\circ$ is elementwise multiplication.

**filter(Q, P) $\rightarrow$ P**  This module masks the input paragraph attention conditioned on the question, selecting a subset of the attended paragraph (e.g., selecting fields goals "in the second

---

[1] We extract numbers and dates as a pre-processing step explained in the Appendix (§A.3)

quarter" in Fig. 1). We compute a *locally-normalized* paragraph-token mask $M \in \mathbb{R}^m$ where $M_j$ is the masking score for the $j$-th paragraph token computed as $M_j = \sigma(\mathbf{w_{filter}}^T[\mathbf{q}\,;\mathbf{P}_{j:}\,;\mathbf{q} \circ \mathbf{P}_{j:}])$. Here $\mathbf{q} = \sum_i Q_i \cdot \mathbf{Q}_{i:} \in \mathbb{R}^d$, is a weighted sum of question-token embeddings, $\mathbf{w_{filter}}^T \in \mathbb{R}^{3d}$ is a learnable parameter vector, and $\sigma$ is the *sigmoid* non-linearity function. The output is a normalized masked input paragraph attention, $P_{\text{filtered}} = \text{normalize}(M \circ P)$.

**relocate(Q, P) $\rightarrow$ P**  This module re-attends to the paragraph based on the question and is used to find the arguments for paragraph spans (e.g., shifting the attention from "field goals" to "who kicked" them). We first compute a paragraph-to-paragraph attention matrix $\mathbf{R} \in \mathbb{R}^{m \times m}$ based on the question, as $\mathbf{R}_{ij} = \mathbf{w_{relocate}}^T[(\mathbf{q} + \mathbf{P}_{i:})\,;\mathbf{P}_{j:}\,;(\mathbf{q} + \mathbf{P}_{i:}) \circ \mathbf{P}_{j:}]$, where $\mathbf{q} = \sum_i Q_i \cdot \mathbf{Q}_{i:} \in \mathbb{R}^d$, and $\mathbf{w_{relocate}} \in \mathbb{R}^{3d}$ is a learnable parameter vector. Each row of $\mathbf{R}$ is also normalized using the softmax operation. The output attention is a weighted sum of the rows $\mathbf{R}$ weighted by the input paragraph attention, $P_{\text{relocated}} = \sum_i P_i \cdot \mathbf{R}_{i:}$

**find-num(P) $\rightarrow$ N**  This module finds a number distribution associated with the input paragraph attention. We use a paragraph token-to-number-token attention map $\mathbf{A}^{\text{num}} \in \mathbb{R}^{m \times N_{\text{tokens}}}$ whose $i$-th row is probability distribution over number-containing tokens for the $i$-th paragraph token. We first compute a token-to-number similarity matrix $\mathbf{S}^{\text{num}} \in \mathbb{R}^{m \times N_{\text{tokens}}}$ as, $\mathbf{S}^{\text{num}}_{i,j} = \mathbf{P}_{i:}^T \mathbf{W}_{\text{num}} \mathbf{P}_{n_j:}$, where $n_j$ is the index of the $j$-th number token and $\mathbf{W}_{\text{num}} \in \mathbb{R}^{d \times d}$ is a learnable parameter. $\mathbf{A}^{\text{num}}_{i:} = \text{softmax}(\mathbf{S}^{\text{num}}_{i:})$. We compute an *expected* distribution over the number tokens $T = \sum_i P_i \cdot \mathbf{A}^{\text{num}}_{i:}$ and aggregate the probabilities for number-tokens with the same value to compute the output distribution $N$. For example, if the values of the number-tokens are [*2, 2, 3, 4*] and $T = [0.1, 0.4, 0.3, 0.2]$, the output will be a distribution over $\{2, 3, 4\}$ with $N = [0.5, 0.3, 0.2]$.

**find-date(P) $\rightarrow$ D** follows the same process as above to compute a distribution over dates for the input paragraph attention. The corresponding learnable parameter matrix is $\mathbf{W}_{\text{date}} \in \mathbb{R}^{d \times d}$.

**count(P) $\rightarrow$ C**  This module is used to count the number of attended paragraph spans. The idea is to learn a module that detects contiguous spans of attention values and counts each as one. For example, if an attention vector is $[0, 0, 0.3, 0.3, 0, 0.4]$, the count module should produce an output of $2$. The module first scales the attention using the values $[1, 2, 5, 10]$ to convert it into a matrix $P_{\text{scaled}} \in \mathbb{R}^{m \times 4}$. A bidirectional-GRU then represents each token attention as a hidden vector $h_t$. A single-layer feed-forward network maps this representation to a soft 0/1 score to indicate the presence of a span surrounding it. These scores are summed to compute a count value, $c_v = \sum \sigma\left(FF(\text{countGRU}(P_{\text{scaled}}))\right) \in \mathbb{R}$. We hypothesize that the output count value is normally distributed with $c_v$ as mean, and a constant variance $v = 0.5$, and compute a categorical distribution over the supported count values, as $p(c) \propto \exp\left(-(c-c_v)^2/2v^2\right) \ \forall c \in [0, 9]$. Pretraining this module by generating synthetic data of attention and count values helps (see §A.4).

**compare-num-lt(P1, P2) $\rightarrow$ P**  This module performs a soft less-than operation between two passage distributions. For example, to find *the city with fewer people, cityA or cityB*, the module would output a linear combination of the two input attentions weighted by which city was associated with a lower number. This module internally calls the `find-num` module to get a number distribution for each of the input paragraph attentions, $N_1$ and $N_2$. It then computes two soft boolean values, $p(N_1 < N_2)$ and $p(N_2 < N_1)$, and outputs a weighted sum of the input paragraph attentions. The boolean values are computed by marginalizing the relevant joint probabilities:

$$p(N_1 < N_2) = \sum_i \sum_j \mathbb{1}_{N_1^i < N_2^j} N_1^i N_2^j \qquad p(N_2 < N_1) = \sum_i \sum_j \mathbb{1}_{N_2^i < N_1^j} N_2^i N_1^j$$

The final output is, $P_{out} = p(N_1 < N_2) * P_1 + p(N_2 < N_1) * P_2$. When the the predicted number distributions are peaky, $p(N_1 < N_2)$ or $p(N_2 < N_1)$ is close to $1$, and the output is either $P_1$ or $P_2$.

We similarly include the comparison modules `compare-num-gt`, `compare-date-lt`, and `compare-date-gt`, defined in an essentially identical manner, but for greater-than and for dates.

**time-diff(P1, P2) $\rightarrow$ TD**  The module outputs the difference between the dates associated with the two paragraph attentions as a distribution over all possible difference values. The module internally calls the `find-date` module to get a date distribution for the two paragraph attentions,

$D_1$ and $D_2$. The probability of the difference being $t_d$ is computed by marginalizing over the joint probability for the dates that yield this value, as $p(t_d) = \sum_{i,j} \mathbb{1}_{(d_i - d_j = t_d)} D_1^i D_2^j$.

**find-max-num(P) $\rightarrow$ P, find-min-num(P) $\rightarrow$ P**   Given a passage attention attending to multiple spans, this module outputs an attention for the span associated with the largest (or smallest) number. We first compute an expected number token distribution $T$ using find-num, then use this to compute the expected probability that each number token is the one with the maximum value, $T^{\max} \in \mathbb{R}^{N_{\text{tokens}}}$ (explained below). We then re-distribute this distribution back to the original passage tokens associated with those numbers. The contribution from the $i$-th paragraph token to the $j$-th number token, $T_j$, was $P_i \cdot \mathbf{A}^{\text{num}}{}_{ij}$. To compute the new attention value for token $i$, we re-weight this contribution based on the ratio $T_j^{\max}/T_j$ and marginalize across the number tokens to get the new token attention value: $\bar{P}_i = \sum_j T_j^{\max}/T_j \cdot P_i \cdot \mathbf{A}^{\text{num}}{}_{ij}$.

**Computing $T^{\mathbf{max}}$**: Consider a distribution over numbers $N$, sorted in an increasing order. Say we sample a set $S$ (size $n$) of numbers from this distribution. The probability that $N_j$ is the largest number in this set is $p(x \leq N_j)^n - p(x \leq N_{j-1})^n$ i.e. all numbers in $S$ are less than or equal to $N_j$, and at least one number is $N_j$. By picking the set size $n = 3$ as a hyperparameter, we can analytically (and differentiably) convert the expected distribution over number tokens, $T$, into a distribution over the maximum value $T^{\max}$.

**span(P) $\rightarrow$ S**   This module is used to convert a paragraph attention into a contiguous answer span and only appears as the outermost module in a program. The module outputs two probability distributions, $P_s$ and $P_e \in \mathbb{R}^m$, denoting the probability of a token being the start and end of a span, respectively. This module is implemented similar to the count module (see §A.5).

## 4   AUXILIARY SUPERVISION

As mentioned in §2.2, jointly learning the parameters of the parser and the modules using only end-task QA supervision is extremely challenging. To overcome issues in learning, (a) we introduce an unsupervised auxiliary loss to provide an inductive bias to the execution of find-num, find-date, and relocate modules (§4.1); and (b) provide heuristically-obtained supervision for question program and intermediate module output (§4.2) for a subset of questions (5–10%).

### 4.1   UNSUPERVISED AUXILIARY LOSS FOR IE

The find-num, find-date, and relocate modules perform information extraction by finding relevant arguments for entities and events mentioned in the context. In our initial experiments we found that these modules would often spuriously predict a high attention score for output tokens that appear far away from their corresponding inputs. We introduce an auxiliary objective to induce the idea that the arguments of a mention should appear near it. For any token, the objective increases the sum of the attention probabilities for output tokens that appear within a window $W = 10$, letting the model distribute the mass within that window however it likes. The objective for the find-num is

$$H_{\text{loss}}^{\text{n}} = -\sum_{i=1}^{m} \log \Big( \sum_{j=0}^{N_{\text{tokens}}} \mathbb{1}_{n_j \in [i \pm W]} \mathbf{A}^{\text{num}}{}_{ij} \Big)$$

We compute a similar loss for the date-attention map $\mathbf{A}^{\text{date}}$ ($H_{\text{loss}}^{\text{d}}$) and the relocate-map $\mathbf{R}$ ($H_{\text{loss}}^{\text{r}}$). The final auxiliary loss is $H_{\text{loss}} = H_{\text{loss}}^{\text{n}} + H_{\text{loss}}^{\text{d}} + H_{\text{loss}}^{\text{r}}$.

### 4.2   QUESTION PARSE AND INTERMEDIATE MODULE OUTPUT SUPERVISION

**Question Parse Supervision.**   Learning to parse questions in a noisy feedback environment is very challenging. For example, even though the questions in CLEVR are programmatically generated, Hu et al. (2017) needed to pre-train their parser using external supervision for all questions. For DROP, we have no such external supervision. In order to bootstrap the parser, we analyze some questions manually and come up with a few heuristic patterns to get program and corresponding question attention supervision (for modules that require it) for a subset of the training data (10%

of the questions; see §A.6). For example, for program `find-num(find-max-num(find()))`, we provide supervision for question tokens to attend to when predicting the `find` module.

**Intermediate Module Output Supervision.** Consider the question, "how many yards was the shortest goal?". The model only gets feedback for how long the *shortest goal* is, but not for other *goals*. Such feedback biases the model in predicting incorrect values for intermediate modules (only the shortest goal instead of all in `find-num`) which in turn hurts model generalization.

We provide heuristically-obtained noisy supervision for the output of the `find-num` and `find-date` modules for a subset of the questions (5%) for which we also provide question program supervision. For questions like "how many yards was the longest/shortest touchdown?", we identify all instances of the token "touchdown" in the paragraph and assume the closest number to it should be an output of the `find-num` module. We supervise this as a multi-hot vector $N^*$ and use an auxiliary loss, similar to question-attention loss, against the output distribution $N$ of `find-num`. We follow the same procedure for a few other question types involving dates and numbers; see §A.7 for details.

## 5 EXPERIMENTS

### 5.1 DATASET

We perform experiments on a portion of the recently released DROP dataset (Dua et al., 2019), which to the best of our knowledge is the only dataset that requires the kind of compositional and symbolic reasoning that our model aims to solve. Our model possesses diverse but limited reasoning capability; hence, we try to automatically extract questions in the scope of our model based on their first n-gram. These n-grams were selected by performing manual analysis on a small set of questions. The dataset we construct contains $20,000$ questions for training/validation, and $1800$ questions for testing ($25\%$ of DROP). Since the DROP test set is hidden, this test set is extracted from the validation data. Though this is a subset of the full DROP dataset it is still a significantly-sized dataset that allows drawing meaningful conclusions. We make our subset and splits available publicly with the code.

Based on the manual analysis we classify these questions into different categories, which are:
**Date-Compare** e.g. *What happened last, commission being granted to Robert or death of his cousin?*
**Date-Difference** e.g. *How many years after his attempted assassination was James II coronated?*
**Number-Compare** e.g. *Were there more of cultivators or main agricultural labourers in Sweden?*
**Extract-Number** e.g. *How many yards was Kasay's shortest field goal during the second half?*
**Count** e.g. *How many touchdowns did the Vikings score in the first half?*
**Extract-Argument** e.g. *Who threw the longest touchdown pass in the first quarter?*

**Auxiliary Supervision** Out of the $20,000$ training questions, we provide question program supervision for $10\%$ (2000), and intermediate module output supervision for $5\%$ (1000) of training questions. We use curriculum learning (Bengio et al., 2009) where the model is trained only on heuristically-supervised non-count questions for the first $5$ epochs.

### 5.2 RESULTS

We compare to publicly available best performing models: NAQANet (Dua et al., 2019), NABERT+ (Kinley & Lin, 2019), TAG-NABERT+ (Avia Efrat & Shoham, 2019), and MTMSN (Hu et al., 2019), all trained on the same data as our model. We implement our model using AllenNLP (Gardner et al., 2018). [2]

The hyperparameters used for our model are described in the appendix. All results are reported as an average of 4 model runs.

**Overall.** Table 2a compares our model's performance to state-of-the-art models on our full test set. Our model achieves an F1 score of 73.1 (w/ GRU) and significantly outperforms NAQANet (62.1 F1). Using BERT representations, our model's performance increases to 77.4 F1 and outperforms SoTA models that use BERT representations, such as MTMSN (76.5 F1). This shows the efficacy of our proposed model in understanding complex compositional questions and performing multi-step

---

[2]Our code is available at `http://cogcomp.org/page/publication_view/899`.

| Model | F1 | EM |
|---|---|---|
| NAQANET | 62.1 | 57.9 |
| TAG-NABERT+ | 74.2 | 70.6 |
| NABERT+ | 75.4 | 72.0 |
| MTMSN | 76.5 | 73.1 |
| OUR MODEL (W/ GRU) | 73.1 | 69.6 |
| OUR MODEL (W/ BERT) | **77.4** | **74.0** |

(a) **Performance on DROP (pruned)**

| Question Type | MTMSN | Our Model (w/ BERT) |
|---|---|---|
| DATE-COMPARE (18.6%) | **85.2** | 82.6 |
| DATE-DIFFERENCE (17.9%) | 72.5 | **75.4** |
| NUMBER-COMPARE (19.3%) | 85.1 | **92.7** |
| EXTRACT-NUMBER (13.5%) | 80.7 | **86.1** |
| COUNT (17.6%) | **61.6** | 55.7 |
| EXTRACT-ARGUMENT (12.8%) | 66.6 | **69.7** |

(b) **Performance by Question Type (F1)**

Table 2: **Performance of different models on the dataset and across different question types**.

reasoning over natural language text. Additionally, this shows that structured models still benefit when used over representations from large pretrained-LMs, such as BERT.

**Performance by Question Type.** Table 2b shows the performance for different question types as identified by our heuristic labeling. Our model outperforms MTMSN on majority of question types but struggles with counting questions; it outperforms MTMSN on only some of the runs. Even after pre-training the count module using synthetic data, training it is particularly unstable. We believe this is because feedback from count questions is weak, i.e., the model only gets feedback about the count value and not what the underlying set is; and because it was challenging to define a categorical count distribution given a passage attention distribution— finding a better way to parameterize this function is an interesting problem for future work.

**Effect of Additional Supervision.** Figure 2a shows that the unsupervised auxiliary objective significantly improves model performance (from 57.3 to 73.1 F1). The model using BERT diverges while training without the auxiliary objective. Additionally, the intermediate module output supervision has slight positive effect on the model performance.

**Effect of Training Data Size.** Figure 2b shows that our model significantly outperforms MTMSN when training using less data, especially using 10-25% of the available supervision. This shows that by explicitly modeling compositionality, our model is able to use additional auxiliary supervision effectively and achieves improved model generalization.

**Incorrect Program Predictions.** Mistakes by our model can be classified into two types; incorrect program prediction and incorrect execution. Here we show few mistakes of the first type that highlight the need to parse the question in a context conditional manner:

1. How many touchdown passes did Tom Brady throw in the season? - `count(find)` is incorrect since the correct answer requires a simple lookup from the paragraph.

2. Which happened last, failed assassination attempt on Lenin, or the Red Terror? - `date-compare-gt(find, find))` is incorrect since the correct answer requires natural language inference about the order of events and not symbolic comparison between dates.

3. Who caught the most touchdown passes? - `relocate(find-max-num(find)))`. Such questions, that require nested counting, are out of scope of our defined modules because the model would first need to to count the passes caught by each player.

## 6 RELATED WORK

Semantic parsing techniques have been used for a long time for compositional question understanding. Approaches have used labeled logical-forms (Zelle & Mooney, 1996; Zettlemoyer & Collins, 2005), or weak QA supervision (Clarke et al., 2010; Berant et al., 2013; Reddy et al., 2014) to learn parsers to answer questions against structured knowledge bases. These have also been extended for QA using symbolic reasoning against semi-structured tables (Pasupat & Liang, 2015; Krishnamurthy et al., 2017; Neelakantan et al., 2016). Recently, BERT-based models for DROP have been been

| Supervision Type | | w/ BERT | w/ GRU |
|:---:|:---:|:---:|:---:|
| $H_{loss}$ | MOD-SUP | | |
| ✓ | ✓ | **77.4** | **73.1** |
| ✓ | | 76.3 | 71.8 |
| | ✓ | –* | 57.3 |

(a) **Effect of Auxiliary Supervision:** The auxiliary loss contributes significantly to the performance, whereas module output supervision has little effect. *Training diverges without $H_{loss}$ for the BERT-based model.*

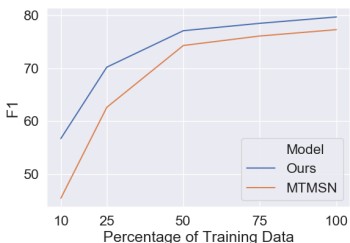

(b) **Performance with less training data**: Our model performs significantly better than the baseline with less training data, showing the efficacy of explicitly modeling compositionality.

Figure 2: Effect of auxiliary losses and the size of training data on model performance.

proposed (Hu et al., 2019; Andor et al., 2019; Kinley & Lin, 2019), but all these models essentially perform a multiclass classification over pre-defined programs. Our model on the other hand provides an interpretable, compositional parse of the question and exposes its intermediate reasoning steps.

For combining learned execution modules with semantic parsing, many variations to NMNs have been proposed; NMN (Andreas et al., 2016) use a PCFG parser to parse the question and only learn module parameters. N2NMNs (Hu et al., 2017) simultaneously learn to parse and execute but require pre-training the parser. Gupta & Lewis (2018) propose a NMN model for QA against knowledge graphs and learn execution for semantic operators from QA supervision alone. Recent works (Gupta & Lewis, 2018; Mao et al., 2019) also use domain-knowledge to alleviate issues in learning by using curriculum learning to train the executor first on simple questions for which parsing is not an issue. All these approaches perform reasoning on synthetic domains, while our model is applied to natural language. Concurrently, Jiang & Bansal (2019) apply NMN to HotpotQA (Yang et al., 2018) but their model comprises of only 3 modules and is not capable of performing symbolic reasoning.

## 7 FUTURE DIRECTIONS

We try a trivial extension to our model by adding a module that allows for addition & subtraction between two paragraph numbers. The resulting model achieves a score of $65.4$ F1 on the complete validation data of DROP, as compared to MTMSN that achieves 72.8 F1.

Manual analysis of predictions reveals that a significant majority of mistakes are due to insufficient reasoning capability in our model and would require designing additional modules. For example, questions such as (a) How many languages each had less than $115,000$ speakers in the population? and Which racial groups are smaller than $2\%$? would require pruning passage spans based on the numerical comparison mentioned in the question; (b) Which quarterback threw the most touchdown passes? and In which quarter did the teams both score the same number of points? would require designing modules that considers some key-value representation of the paragraph; (c) How many points did the packers fall behind during the game? would require IE for implicit argument (points scored by the other team). It is not always clear how to design interpretable modules for certain operations; for example, for the last two cases above.

It is worth emphasizing here what happens when we try to train our model on these questions for which our modules *can't* express the correct reasoning. The modules in the predicted program get updated to try to perform the reasoning anyway, which harms their ability to execute their intended operations (cf. §2.2). This is why we focus on only a subset of the data when training our model.

In part due to this training problem, some other mistakes of our model relative to MTMSN on the full dataset are due to incorrect execution of the intermediate modules. For example, incorrect grounding by the find module, or incorrect argument extraction by the find-num module. For mistakes such as these, our NMN based approach allows for identifying the cause of mistakes and supervising these modules using additional auxiliary supervision that is not possible in black-box models. This additionally opens up avenues for transfer learning where modules can be independently trained

using indirect or distant supervision from different tasks. Direct transfer of reasoning capability in black-box models is not so straight-forward.

To solve both of these classes of errors, one could use black-box models, which gain performance on some questions at the expense of limited interpretability. It is not trivial to combine the two approaches, however. Allowing black-box operations inside of a neural module network significantly harms the interpretability—e.g., an operation that directly answers a question after an encoder, mimicking BERT-QA-style models, encourages the encoder to perform complex reasoning in a non-interpretable way. This also harms the ability of the model to use the interpretable modules even when they would be sufficient to answer the question. Additionally, due to our lack of supervised programs, training the network to use the interpretable modules instead of a black-box shortcut module is challenging, further compounding the issue. Combining these black-box operations with the interpretable modules that we have presented is an interesting and important challenge for future work.

## 8  CONCLUSION

We show how to use neural module networks to answer compositional questions requiring symbolic reasoning against natural language text. We define probabilistic modules that propagate uncertainty about symbolic reasoning operations in a way that is end-to-end differentiable. Additionally, we show that injecting inductive bias using unsupervised auxiliary losses significantly helps learning.

While we have demonstrated marked success in broadening the scope of neural modules and applying them to open-domain text, it remains a significant challenge to extend these models to the full range of reasoning required even just for the DROP dataset. NMNs provide interpretability, compositionality, and improved generalizability, but at the cost of restricted expressivity as compared to more black box models. Future research is necessary to continue bridging these reasoning gaps.

ACKNOWLEDGMENTS

We would like to thank Daniel Deutsch and the anonymous reviewers for their helpful comments. This material is based upon work sponsored in part by the DARPA MCS program under Contract No. N660011924033 with the United States Office Of Naval Research, an ONR award, the LwLL DARPA program, and a grant from AI2.

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

# A   APPENDIX

## A.1   QUESTION AND PARAGRAPH ENCODER

Our model represents the question $q$ as $\mathbf{Q} \in \mathbb{R}^{n \times d}$ and paragraph $p$ as $\mathbf{P} \in \mathbb{R}^{m \times d}$ using contextualized token embeddings. These embeddings are either produced using a multi-layer GRU network that is trained from scratch, or a pr-trained BERT model that is fine-tuned during training.

**GRU**: We use a 2-layer, 64-dimensional ($d = 128$, effectively), bi-directional GRU. The same GRU is used for both, the question and the paragraph. The token embeddings input to the contextual encoder are a concatenation of 100-d pre-trained GloVe embeddings, and 200-d embeddings output from a CNN over the token's characters. The CNN uses filters of size=5 and character embeddings of 64-d. The pre-trained glove embeddings are fixed, but the character embeddings and the parameters for the CNN are jointly learned with the rest of the model.

**BERT**: The input to the BERT model is the concatenation of the question and paragraph in the following format: `[CLS] Question [SEP] Context [SEP]`. The question and context tokens input to the BERT model are sub-words extracted by using BERT's tokenizer. We separate the question and context representation from the output of BERT as $\mathbf{Q}$ and $\mathbf{P}$, respectively. We use 'bert-base-uncased' model for all out experiments.

## A.2   QUESTION PARSER DECODER

The decoder for question parsing is a single-layer, 100-dimensional, LSTM. For each module, we use a 100-dimensional embedding to present it as an action in the decoder's input/output vocabulary. The attention is computed as a dot-product between the decoder hidden-state and the encoders hidden states which is normalized using the softmax operation.

As the memory-state for the zero-eth time-step in the decoder, we use the last hidden-state of the question encoder GRU, or the `[CLS]` embedding for the BERT-based model.

We use a beam-size of $4$ for the approximate maximum marginal likelihood objective. Optmization is performed using the Adam algorithm with a learning rate of $0.001$ or using BERT's optimizer with a learning rate of $1e - 5$.

## A.3   NUMBER AND DATE PARSING

We pre-process the paragraphs to extract the numbers and dates in them. For numbers, we use a simple strategy where all tokens in the paragraph that can be parsed as a number are extracted. For example, 200 in "200 women". The total number of number-tokens in the paragraph is denoted by $N_{\text{tokens}}$. We do not normalize numbers based on their units and leave it for future work.

To extract dates from the paragraph, we run the spaCy-NER[3] and collect all "DATE" mentions. To normalize the date mentions we use an off-the-shelf date-parser[4]. For example, a date mention "19th November, 1961" would be normalized to $(19, 11, 1961)$ (day, month, year). The total number of date-tokens is denoted by $D_{\text{tokens}}$

## A.4   PRE-TRAINING COUNT MODULE

As mentioned in the paper, training the `count` module is challenging and found that pre-training the parameters of the `count` module helps.

To re-iterate, the module gets as input a paragraph attention $P \in \mathbb{R}^m$. The module first scales the attention using the values $[1, 2, 5, 10]$ to convert it into a matrix $P_{\text{scaled}} \in \mathbb{R}^{m \times 4}$. A bidirectional-GRU then represents each token attention as a hidden vector $h_t$. A single-layer feed-forward network maps this representation to a soft 0/1 score to indicate the presence of a span surrounding it. These scores

---

[3]https://spacy.io/
[4]https://github.com/scrapinghub/dateparser

are summed to compute a count value, $c_v$.

$$\text{count}_{\text{scores}} = \sigma\Big(FF(\text{countGRU}(P_{\text{scaled}}))\Big) \in \mathbb{R}^m$$

$$c_v = \sum \text{count}_{\text{scores}} \in \mathbb{R}$$

We generate synthetic data to pre-train this module; each instance is a normalized-attention vector $x = \mathbb{R}^m$ and a count value $y \in [0, 9]$. This is generated by sampling $m$ uniformly between $200 - 600$, then sampling a count value $y$ uniformly in $[0, 9]$. We then sample $y$ span-lengths between $5 - 15$ and also sample $y$ non-overlapping span-positions in the attention vector $x$. For all these $y$ spans in $x$, we put a value of $1.0$ and zeros everywhere else. We then add 0-mean, 0.01-variance gaussian noise to all elements in $x$ and normalize to make the normalized attention vector that can be input to the count module.

We train the parameters of the count module using these generated instances using $L_2$-loss between the true count value and the predicted $c_v$.

The countGRU in the `count` module (spanGRU – `span` module) is a 2-layer, bi-directional GRU with input-dim = 4 and output-dim = 20. The final feed-forward comprises of a single-layer to map the output of the countGRU into a scalar score.

## A.5 SPAN MODULE

The `span` module is implemented similar to the `count` module. The input paragraph attention is first scaled using $[1, 2, 5, 10]$, then a bidirectional-GRU represents each attention as a hidden vector, and a single-layer feed-forward network maps this to 2 scores, for span start and end. A softmax operation on these scores gives the output probabilities.

## A.6 AUXILIARY QUESTION PARSE SUPERVISION

For questions with parse supervision $\mathbf{z}^*$, we decouple the marginal likelihood into two maximum likelihood objectives, $p(\mathbf{z}^*|q)$ and $p(y^*|\mathbf{z}^*)$. We also add a loss for the decoder to attend to the tokens in the question attention supervision when predicting the relevant modules. The question attention supervision is provided as a mutli-hot vector $\alpha^* \in \{0, 1\}^n$. The loss against the predicted attention vector $\alpha$ is, $Q_{\text{loss}} = -\sum_{i=1}^{n} \alpha_i^* \log \alpha_i$. Since the predicted attention is a normalized distribution, the objective increases the sum of log-probabilities of the tokens in the supervision.

The following patterns are used to extract the question parse supervision for the training data:

1. *what happened first SPAN1 or SPAN2?*
   `span(compare-date-lt(find(), find()))`: with `find` attentions on SPAN1 and SPAN2, respectively. Use `compare-date-gt`, if *second* instead of *first*.

2. *were there fewer SPAN1 or SPAN2?*
   `span(compare-num-lt(find(), find()))`: with `find` attentions on SPAN1 and SPAN2, respectively. Use `compare-num-gt`, if *more* instead of *fewer*.

3. *how many yards was the longest {touchdown / field goal}?*
   `find-num(find-max-num(find()))`: with `find` attention on *touchdown / field goal*. For *shortest*, the `find-min-num` module is used.

4. *how many yards was the longest {touchdown / field goal} SPAN ?*
   `find-num(find-max-num(filter(find())))`: with `find` attention on *touchdown / field goal* and filter attention on all SPAN tokens.

5. *how many {field goals, touchdowns, passes} were scored SPAN?*
   `count(filter(find()))`: with `find` attention on {*field goals, touchdowns, passes*} and `filter` attention on SPAN.

6. *who {kicked, caught, threw, scored} SPAN?*
   `span(relocate(filter(find())))`: with `relocate` attention on {*kicked, caught, threw, scored*}, `find` attention on {*touchdown / field goal*}, and `filter` attention on all other tokens in the SPAN.

> Question: Which group was smaller, the 25 to 44 year olds or the 45 to 64 year olds?
>
> Program: `span(compare-num-lt(find, find))`
>
> Answer: 45 to 64
>
> *compare-num-lt passage-attention:* In the city, the population was spread out with 12.0% under the age of 18, 55.2% from 18 to 24, 15.3% from 25 to 44, 10.3% from 45 to 64, and 7.1% who were 65 years of age or older. The median age was 22 years. For every 100 females, there were 160.7 males. For every 100 females age 18 and over, there were 173.2 males.
>
> *number-distribution*: 7.1 10.3 12 15.3 18 22 24 25 44 45 55.2 64 65 100 160.7 173.2
>
> *question-attention*: Which group was smaller, the 25 to 44 year olds or the 45 to 64 year olds?
>
> *passage-attention:* In the city, the population was spread out with 12.0% under the age of 18, 55.2% from 18 to 24, 15.3% from 25 to 44, 10.3% from 45 to 64, and 7.1% who were 65 years of age or older. The median age was 22 years. For every 100 females, there were 160.7 males. For every 100 females age 18 and over, there were 173.2 males.

Figure 3: Example usage of `num-compare-lt`: Our model predicts the program `span(compare-num-lt(find, find))` for the given question. We show the question attentions and the predicted passage attentions of the two `find` operations using color-coded highlights on the same question and paragraph (to save space) at the bottom. The number grounding for the two paragraph attentions predicted in the `compare-num-lt` module are shown using the same colors in *number-distribution*. Since the number associated to the passage span "45 to 64" is lower (10.3 vs. 15.3), the output of the `compare-num-lt` module is "45 to 64" as shown in the passage above.

## A.7 HEURISTIC INTERMEDIATE MODULE OUTPUT SUPERVISION

As mentioned in Section 4.3, we heuristically find supervision for the output of the `find-num` and `find-date` module for a subset of questions that already contain question program supervision. These are as follows:

1. *how many yards was the longest/shortest {touchdown, field goal}?*
   We find all instances of touchdown/field goal in the passage and assume that the number appearing closest should be an output of the `find-num` module.

2. *what happened first EVENT1 or EVENT2?*
   Similar to above, we perform fuzzy matching to find the instance of EVENT1 and EVENT2 in the paragraph and assume that the closest dates should be the output of the two `find-date` module calls made by the `compare-date-lt` module in the gold program.

3. *were there fewer SPAN1 or SPAN2?*
   This is exactly the same as previous for `find-num` module calls by `compare-num-lt`.

## A.8 EXAMPLE PREDICTIONS

In Figures 3, 4, 5, 6, 7 we show predictions by our model that shows the learned execution of various modules defined in the paper.

Question: Which event happened first, when king Chulalongkorn enacted two decrees banning the capture and sale of Kha slaves or when Siam abolished the tributes collected from vassal states?

Program: span(compare-date-lt(find, find))

Answer: banning the capture and sale of Kha slaves

*date-distribution:* 1874/-1/-1 1884/-1/-1 1868/-1/-1 1883/-1/-1 1899/-1/-1

*question-attention*: Which event happened first, when king Chulalongkorn enacted two decrees banning the capture and sale of Kha slaves or when Siam abolished the tributes collected from vassal states?

*passage-attention:* Before the Monthon reforms initiated by king Chulalongkorn, Siamese territories were divided into three categories: Inner Provinces forming the core of the kingdom, Outer Provinces that were adjacent to the inner provinces and tributary states located on the border regions. The area of southern Laos that came under Siamese control following the Lao rebellion and destruction of Vientiane belonged to the later category, maintaining relative autonomy. Lao nobles who had received the approval of the Siamese king exercised authority on the Lao population as well as the Alak and Laven-speaking tribesmen. Larger tribal groups often raided weaker tribes abducting people and selling them into slavery at the trading hub of Champasak, while themselves falling prey to Khmer, Lao and Siamese slavers. From Champasak the slaves were transported to Phnom-Penh and Bangkok, thus creating a large profits for the slavers and various middlemen. In 1874 and 1884, king Chulalongkorn enacted two decrees banning the capture and sale of Kha slaves while also freeing all slaves born after 1868. Those abolitionist policies had an immediate effect on slave trading communities. In 1883, France attempted to expand its control in Southeast Asia by claiming that the Treaty of Huế extended into all Vietnamese vassal states. French troops gradually occupied the Kontum Plateau and pushed the Siamese from Laos following the Franco-Siamese War. A new buffer zone was thus created on the west bank of Mekong, as the area lacked the presence of the Siamese military local outlaws flocked the newly created safe haven. In 1899, Siam abolished the tributes collected from vassal states, replacing them with a new tax collected from all able bodied men, undermining the authority of Lao officials.

Figure 4: Example usage of `date-compare-lt`: Similar to Fig. 3, we show the question attentions, the output passage attentions of the `find` module, and the date grounding predicted in the `compare-date-lt` module in color-coded highlights. The passage span predicted as the answer is the one associated to a lower-valued date.

Question: How many rushing touchdowns were scored in the game?

Program: count(find)

Answer: 5

*count-distribution:* 0 1 2 3 4 5 6 7 8 9

*question-attention*: How many rushing touchdowns were scored in the game?

*passage-attention:* Coming off their Thursday night home win over the Packers, the Cowboys flew to Ford Field for a Week 14 interconference duel with the Detroit Lions. In the first quarter, Dallas trailed early as Lions RB T.J. Duckett getting a 32-yard TD run, along with kicker Jason Hanson getting a 19-yard field goal. In the second quarter, the Cowboys got on the board with RB Marion Barber getting a 20-yard TD run. Detroit would answer with Hanson kicking a 36-yard field goal, while RB Kevin Jones getting a 2-yard TD run. The Cowboys ended the half with QB Tony Romo completing an 8-yard TD pass to Barber. In the third quarter, the Lions replied with Jones getting a 3-yard TD run for the only score of the period. In the fourth quarter, Dallas came back and took the lead with Barber getting a 1-yard TD run and Romo completing a 16-yard TD pass to TE Jason Witten. With the win, the Cowboys improved to 12-1 and clinched the NFC East crown for the first time since 1998.

Figure 5: Example usage of `count`: For the predicted program `count(find)`, the `find` module predicts all "rushing touchdown" spans from the passage and the `count` module counts their number and represents its output as a distribution over possible count values. In this example, the predicted distribution as a mode of "5".

Question: How many yards was the longest field goal?

Program: `find-num(find-max-num(find))`

Answer: 39

*max-num-distribution*:  1 2 10 11 12 13 29 31 38 **39** 40 1986 1990

*input-num-distribution*: 1 2 10 11 12 13 29 31 38 39 40 1986 1990

*find-max-num passage-attention:* Coming off their road win over the Cardinals, the Giants flew to FedEx Field for a Week 13 NFC East rematch with the Washington Redskins. In the first quarter, New York scored first as QB Eli Manning completed a 40-yard TD pass to WR Amani Toomer, along with kicker John Carney getting a 31-yard field goal. In the second quarter, the Giants increased their lead as Carney got a 38-yard field goal. The Redskins would close out the half with WR Devin Thomas getting a 29-yard TD run. In the third quarter, New York began to pull away as RB Brandon Jacobs got a 1-yard TD run. In the fourth quarter, the Giants sealed the deal with Carney connecting on a 39-yard field goal. With the season-sweep, the Giants improved to 11-1, exceeding the 1986 and 1990 teams (both started 10-2 and eventually won the Super Bowl) for the best 12-game record in franchise history.

*question-attention*: How many yards was the longest field goal?

*passage-attention:* Coming off their road win over the Cardinals, the Giants flew to FedEx Field for a Week 13 NFC East rematch with the Washington Redskins. In the first quarter, New York scored first as QB Eli Manning completed a 40-yard TD pass to WR Amani Toomer, along with kicker John Carney getting a 31-yard field goal. In the second quarter, the Giants increased their lead as Carney got a 38-yard field goal. The Redskins would close out the half with WR Devin Thomas getting a 29-yard TD run. In the third quarter, New York began to pull away as RB Brandon Jacobs got a 1-yard TD run. In the fourth quarter, the Giants sealed the deal with Carney connecting on a 39-yard field goal. With the season-sweep, the Giants improved to 11-1, exceeding the 1986 and 1990 teams (both started 10-2 and eventually won the Super Bowl) for the best 12-game record in franchise history.

Figure 6: Example usage of `find-max-num`: For the predicted program, the model first grounds the question span "field goal" to all field goals in the passage, shown as attention in the bottom half. The `find-max-num` first finds the number associated with the input passage spans (shown as input-num-distribution), then finds the distribution over max value (shown as max-num-distribution), and finally outputs the passage span associated with this max value (shown in the passage attention on top). The `find-num` module finally extracts the number associated with this passage attention as the answer.

Question: Which player scored the longest rushing TD?

Program: `span(relocate(find-max-num(find)))`

Answer: T.J. Duckett

*relocate-question-attention:* Which player scored the longest rushing TD?

*relocate / find-max-num passage-attention:* Coming off their Thursday night home win over the Packers, the Cowboys flew to Ford Field for a Week 14 interconference duel with the Detroit Lions. In the first quarter, Dallas trailed early as Lions RB T.J. Duckett getting a 32-yard TD run, along with kicker Jason Hanson getting a 19-yard field goal. In the second quarter, the Cowboys got on the board with RB Marion Barber getting a 20-yard TD run. Detroit would answer with Hanson kicking a 36-yard field goal, while RB Kevin Jones getting a 2-yard TD run. The Cowboys ended the half with QB Tony Romo completing an 8-yard TD pass to Barber. In the third quarter, the Lions replied with Jones getting a 3-yard TD run for the only score of the period. In the fourth quarter, Dallas came back and took the lead with Barber getting a 1-yard TD run and Romo completing a 16-yard TD pass to TE Jason Witten. With the win, the Cowboys improved to 12-1 and clinched the NFC East crown for the first time since 1998.

*max-num-distribution:* 1 2 3 8 12 14 16 19 20 32 36 1998

*input-num-distribution:* 1 2 3 8 12 14 16 19 20 32 36 1998

*question-attention:* Which player scored the longest rushing TD?

*passage_attention:* Coming off their Thursday night home win over the Packers, the Cowboys flew to Ford Field for a Week 14 interconference duel with the Detroit Lions. In the first quarter, Dallas trailed early as Lions RB T.J. Duckett getting a 32-yard TD run, along with kicker Jason Hanson getting a 19-yard field goal. In the second quarter, the Cowboys got on the board with RB Marion Barber getting a 20-yard TD run. Detroit would answer with Hanson kicking a 36-yard field goal, while RB Kevin Jones getting a 2-yard TD run. The Cowboys ended the half with QB Tony Romo completing an 8-yard TD pass to Barber. In the third quarter, the Lions replied with Jones getting a 3-yard TD run for the only score of the period. In the fourth quarter, Dallas came back and took the lead with Barber getting a 1-yard TD run and Romo completing a 16-yard TD pass to TE Jason Witten. With the win, the Cowboys improved to 12-1 and clinched the NFC East crown for the first time since 1998.

Figure 7: Example usage of `relocate`: Similar to the sub-program `find-max-num(find)` in Fig. 6, the `find` module finds all mentions of "rushing TD" in the passage and `find-max-num` selects the one associated to the largest number. This is the span "getting a 32-yard TD run" in the passage above. The question attention predicted for the `relocate` module and its output passage attention is also shown in the passage above.

