# OpenReview forum: "Neural Module Networks for Reasoning over Text"
_ICLR.cc/2020/Conference — Accept (Poster)_

### Official Review · AnonReviewer1 · 2019-10-23
**Official Blind Review #1**

**Rating:** 6

**Review:**

This paper proposes a model and a training framework for question answering which requires compositional reasoning over  the input text, by building executable neural modules and training based on additional auxiliary supervision signals.

I really like this paper and the approach taken: tackling complex QA tasks is an important topic and current state-of-the-art methods rely heavily on lexical similarities as mentioned in the paper. I think learning differentiable programs is a good direction to address this space.

However in IMHO, the paper as it stands is premature for publication, the primary reason being the lack of strong experimental evidence that where the strength of this approach is compared to other methods compared. To be specific, the results in Table 2 are very close between MTMSN and the BERT-based model proposed and it's not clear if the difference is because (1) the model is generally better; (2) this is a subset of the dataset that this model performs better; (3) this is because of the additional supervision signals provided (e.g. the results of Fig 2a without the aux-sup is almost the same as MTMSN) and if we provided similar auxiliary supervision for other models they would equally do well; (4) due to lack of reporting variance and error-bars across runs we see a small increase which may not be significant; ...

Again, the paper is very interesting, but I don't think it's clear and thorough to experimentally prove that the overall approach is working better.


**Experience Assessment:**

I have read many papers in this area.

**Review Assessment: Checking Correctness Of Derivations And Theory:**

I carefully checked the derivations and theory.

**Review Assessment: Checking Correctness Of Experiments:**

I carefully checked the experiments.

**Review Assessment: Thoroughness In Paper Reading:**

I read the paper thoroughly.

---

> ### Author Response · Authors · 2019-11-12
> **Response to AnonReviewer1**
>
> Thank you for your review. While we do agree that our model outperforms MTMSN by a small margin on our complete dev set, we see in Table 2b that NMNs are better on all question types except two: count and date-comparison. Our count module has issues in learning, and date comparison questions in the dataset contain biases that make it likely that a powerful model can figure out the answers without needing to do the discrete reasoning that's modeled by the NMNs. Also, a non trivial proportion of date-comparison questions do not require symbolic reasoning and instead natural language inference (see sec 5.2 -- Incorrect Program Predictions). On all other question types, especially those that most heavily rely on discrete reasoning, our model performs substantially better, because it can explicitly model the reasoning involved.
>
> We performed Welch’s t-test on the different question-type subsets and found out that all differences in performance are statistically significant with a p-value of 0.05 or lower.
>
> Additionally, our model provides a much greater level of interpretability as compared to black-box models like MTMSN. In Appendix A.8 we’ve added examples of predictions from our model that show the level of explanation our model provides.
>
> We do provide additional auxiliary supervision to constrain the latent decisions of our model but that is only possible due to the explicit compositional modeling NMNs perform. It is unclear how to provide such auxiliary supervision to black-box models. In Figure 2(b) we do see that this also helps in achieving better performance with less training data.
>
> Overall, we believe that an approach such as ours, that explicitly models compositionality and is interpretable, is a promising direction to tackle complex multi-step reasoning in NLP. In this paper we lay out the challenges in such modeling choices, provide possible solutions, and show that on a challenging subset of DROP the resulting model performs competitively.

---

### Official Review · AnonReviewer3 · 2019-10-24
**Official Blind Review #3**

**Rating:** 8

**Review:**

Paper Claims

The paper offers a new deep learning approach to symbolic reasoning over text. They propose using Neural Module Networks to perform explicit reasoning steps that are nevertheless differentiable. The process is separated into a semantic parsing of the question, and a resolution using MNMs. Auxiliary tasks improve performance and enable using a BERT pretrained model as a seed. The proposed model's performance surpasses previous SOTA on several question types.

Decision

I'm in favor of accepting this paper because it tackles an extremely challenging and important problem in a novel and successful way. Reasoning has progressed more slowly than other NLP domains, and answering complex multi-reasoning-step questions is a good way to tackle the core of the problem. Regarding the approach, I find the mix of explicit reasoning steps, deep learning modeling, and even heuristics for some of the data preparation, powerful and effective. I see this as a useful step to advance the body of work in this space -- despite not being the desired end-result, it is nevertheless very instructive of what might work for at least some parts of the larger, AI-complete reasoning problem. Also, the paper is clear, well-written, and well-motivated.

Further details on Decision

I'm more than satisfied with the breadth of question types tackled here, and correspondingly the variety of modules. There's extensive, valuable work in designing these modules and making them work together. As the authors point out, it appears that other types of questions will require more intricate modules (or some other means), and I suspect that predetermined modules will not be what generalizes eventually. Nevertheless, the NMN approach taken here can be a stepping stone to further understanding how to tackle symbolic reasoning in a deep neural network. It will be instructive in designing a more ambitious, generalizable model.

The auxiliary supervision tasks appear to be essential to obtaining the results, most notably the unsupervised loss for IE. I think this area has room for further improvement, but what is achieved in the paper is sufficient for publication. In particular, the writing of heuristics is a very specific solution targeting specific types of question and this will not scale to the full scope of natural language questions, and much less to all reasoning. Discussion of how to expand on them, scale them (automatic discovery, some other means?), etc. would be very welcome, as it is the main weakness of the paper.

I also think the methodology is sound and the results are obtained in a reasonable and mostly reproducible way.

This is truly great work that deserves to be published, discussed, and expanded upon.




**Experience Assessment:**

I have read many papers in this area.

**Review Assessment: Checking Correctness Of Derivations And Theory:**

I assessed the sensibility of the derivations and theory.

**Review Assessment: Checking Correctness Of Experiments:**

I assessed the sensibility of the experiments.

**Review Assessment: Thoroughness In Paper Reading:**

I read the paper at least twice and used my best judgement in assessing the paper.

---

> ### Author Response · Authors · 2019-11-12
> **Response to AnonReviewer3**
>
> Thank you for your comments and appreciating our contributions. We completely agree with your evaluation that the exact model we propose in this paper is not going to solve all of the challenges in reasoning but the ideas presented here should benefit future work.
>
> We agree that additional avenues of auxiliary supervision need to be explored for the success of NMNs. Additionally, these do not necessarily have to come from QA supervision. One possible direction is to distantly supervise IE modules from noisy alignments between text and knowledge graphs, or supervise “find” like modules using entity-linking or other grounding supervision. Another one is to create paired questions that share substructures in the reasoning (even if we don’t have the correct answer for one of the questions), and supervise consistency on the shared substructures, which might help the model avoid shortcutting the intended reasoning (e.g., doing the “longest” operation inside of the encoder, or inside of “find”).  Future work should explore these directions that allow for better transfer of modules and supervision.

---

### Official Review · AnonReviewer2 · 2019-10-29
**Official Blind Review #2**

**Rating:** 6

**Review:**

This works applies neural module network to reading comprehension that requires symbolic reasoning. There are two main contributions: (1) the authors designed a set of differentiable neural modules for different operations (for example, arithmetics, sorting, and counting) that is required to perform reasoning over a paragraph of text. These modules can be compositionally combined to perform complex reasoning. And the parameters of each module (which can be viewed as executor of each operation) are learned jointly with the parser that generates thee program composed of those modules. (2) To overcome the challenge of weak supervision, the authors proposed to use auxiliary loss (information extraction loss, parser supervision, intermediate output supervision). The model is evaluated on a subset of DROP, and outperforms the state-of-the-art models. Ablation studies supported the importance of the auxiliary losses.

Strength:

(1) The problem of applying symbolic reasoning over text is important and very challenging. This work has explored a promising direction that applies NMN, which achieved good results in VQA, to QA tasks that requires reasoning, specifically, a subset of the DROP dataset.

(2) The result, although preliminary, seems promising. The design of the modules seems intuitive and the introduction of auxiliary tasks to alleviate the problem of weak supervision is well motivated and works reasonably well.

I am leaning towards rejection because:

(1) The main concern is that the paper, in its current form, seems incomplete. It is understandable that the type of datasets that requires reasoning is not very common nowadays, so only DROP is used for evaluation. However, the current evaluation is only on a subset of DROP, which seems unsatisfying.

The paper argues that "Our model possesses diverse but limited reasoning capability; hence, we try to automatically extract questions in the scope of our model based on their first n-gram". However, results on the full dataset seems necessary for evaluating the potential of NMN approach over text. Even if the result is negative, it is still good to know the cause of the failure. For example, does the difficulty come from unstable training or does it come from insufficient coverage of the modules.

(2) There are several modules introduced in the paper, but there isn't much analysis of them during the experiments. For example, what are some good and bad samples that uses each type of operations.

(3) Since the modules are learned jointly with the parser, it is good to check whether the learned modules are indeed performing the intended operation instead of just adding more capacity to the model. For example, it might help to show a few examples that demonstrates the "compare-num-lt" is actually performing the comparisons. This can support the interpretability claim of the proposed model.


Minor issues:

The complexities of some modules seem large. For example, "compare-num-lt" needs to enumerate all the pairs of numbers, which is quadratic. And the complexity of "find-max-num" depends on the choice of n, which could be large (although it is chosen to be 3 in this work).

It is stated that "Our model performs significantly better than the baseline with less training data, showing the efficacy of explicitly modeling compositionality." However, the comparison with MTMSN using less training data seems a bit unfair since the proposed model is given more supervision (question parse supervision and intermediate module output supervision). Maybe a better argument is that by explicitly modeling compositionality, it is easier to add such extra supervisions than black box models like MTMSN.

For "count", why is the attention scaled using values [1, 2, 5, 10] first?

In summary, I do like the main idea and the paper has merits, but it requires more evaluation and analysis to be accepted. I am willing to increase my score if more contents are added and I look forward to seeing it in a more complete form.

===================================

Update after author response:

Thanks for the clarification and adding the content, I have updated my score accordingly. However, I still believe the impact of this paper will be much larger if the evaluation can be more complete, e.g., evaluating over the full DROP dataset or even include some other datasets. In the current form, it looks borderline.

Selecting a subset (~22.7% of DROP dataset) based on the design of the proposed model ("heuristically chosen based on their first n-gram such that they are covered by our designed modules"), and compare to other models, which can actually handle a broader set of questions, only on the selected subset seems incomplete and raises concerns about how generally applicable the proposed model is. For example, since the proposed model is handling some types of questions better, it would be good to show that it can be combined with other models to get a better overall result.

**Experience Assessment:**

I have published one or two papers in this area.

**Review Assessment: Checking Correctness Of Derivations And Theory:**

N/A

**Review Assessment: Checking Correctness Of Experiments:**

I carefully checked the experiments.

**Review Assessment: Thoroughness In Paper Reading:**

I read the paper thoroughly.

---

> ### Author Response · Authors · 2019-11-12
> **Response to AnonReviewer2**
>
> Thank you for your review and suggestions on how to improve the paper.
>
> (1) While we do agree that the current evaluation is on a subset of DROP; this subset poses a variety of reasoning challenges and is still large enough (~22k questions) to draw scientific conclusions. Is there a particular reason that this evaluation is not enough to claim that NMN based approaches for reasoning over text are a promising direction?
>
> To gauge the challenges in extending the model to even more diverse reasoning challenges, we add a module to perform addition / subtraction between two numbers and achieve 65.4 F1 over the complete validation set for DROP. In comparison MTMSN achieves 72.8 F1.
> On manual analysis we find that a significant majority of mistakes are due to insufficient modules present in our model. For example, questions such as
> - “How many languages each had less than 115,000 speakers in the population?” and “Which racial groups are smaller than 2%?” -- require pruning passage spans based on the numerical comparison mentioned in the question;
> - “Which quarterback threw the most touchdown passes?” and “In which quarter did the teams both score the same number of points?” -- require designing modules that consider some key-value representations for paragraphs
> - “How many points did the packers fall behind during the game?” would require IE for implicit argument (points scored by the other team).
> While sometimes it is clear how to design interpretable modules, some operations, such as the last two cases above, pose challenges that future work should explore. In such cases, black-box models might fare well at the cost of interpretability.
>
> Some other mistakes are caused due to incorrect execution of modules. For example, the find module incorrectly grounding the question span to the passage, or the find-argument module extracting the incorrect argument. For mistakes such as these, our NMNs based approach allows for auxiliary supervision of latent decisions and opens up possibilities of transfer learning that is not obvious in end-to-end black-box models.
>
> We have added the above discussion to a new section 7 in the paper.
>
> (2) We’ve added a few example mistakes our model makes in correctly predicting the program in the sec 5.2 -- Incorrect Program Predictions.
>
> (3) In the Appendix A.8 we add example predictions from our model which shows execution for various modules. To be clear, “compare-num-lt” is a parameter-free module which deterministically outputs the expected probability that N1<N2. However, examples in A.8 show that the modules that have parameters, such as find, find-num, and relocate, do learn to perform the operations that they are designed for.
>
> Minor issues:
> 1) “compare-num-lt” and other similar modules do perform marginalization over pairs of numbers but this is performed as an efficient matrix multiplication. As such modules are applied over very large number support, future work could explore methods that perform approximate marginalization.
> “find-max-num” computes the distribution over the maximum value using the closed form analytical formula: $p(x \leq N_{j})^{n} - p(x \leq N_{j - 1})^{n}$, and hence the complexity does not depend on the hyper-parameter “n”.
>
> 2) We agree that our model’s performance should be attributed to the fact that explicit modeling of compositionality allows for additional auxiliary supervision. There are also interesting insights to gain about the challenge of learning interpretable, compositional models.  It is very easy for the model to shortcut the interpretability that you try to put into it (e.g., hijacking the “find” module to do something other than “find”), and additional supervision is required to combat this.  We have updated the paper to reflect this.  See section 7 and updated wording in section 5.2 (Effect of Training Data Size).
>
> 3) The “count” module gets as input a normalized-passage-attention where passage lengths are typically 400-500 tokens. During our initial experiments we realized that scaling the attention using values >1 helps the model in differentiating amongst small values.

---

### Author Response · Authors · 2019-11-15
**Any final clarifications?**

Since there are only a few hours remaining for the author response period, we wanted to ask if there are any final clarifications that we can provide.

---

### Author Response · Authors · 2020-02-04
**Code release**

We've released the code for this paper here: https://nitishgupta.github.io/nmn-drop

---

### Decision · Program_Chairs · 2019-12-19

**Decision:**

Accept (Poster)

**Comment:**

This work extends the previously introduced NMN for VQA for handling reasoning over text using symbolic reasoning components that can perform counting, sorting etc and can be compositionally combined. Moreover, to successfully train the model, the authors introduce a simple unsupervised auxiliary loss for training the IE components as well heuristically incorporating inductive biases in the behaviour on couple of components. All reviews agreed that this is a challenging topic and an interesting approach to symbolic reasoning over text. At the same time, reviewers did point that experiments are borderline thin, since the authors start with DROP and drop questions that are not particularly suited for symbolic reasoning, resulting in a substantially smaller dataset. Despite the fact that the experiments could probably be stronger, I’m recommending acceptance cause this topic is very interesting and this is a good paper to raise discussions at ICLR,